# Learning Attentional Communication for Multi-Agent Cooperation

**Jiechuan Jiang**
Peking University
jiechuan.jiang@pku.edu.cn

**Zongqing Lu***
Peking University
zongqing.lu@pku.edu.cn

## Abstract

Communication could potentially be an effective way for multi-agent cooperation. However, information sharing among all agents or in predefined communication architectures that existing methods adopt can be problematic. When there is a large number of agents, agents cannot differentiate valuable information that helps cooperative decision making from globally shared information. Therefore, communication barely helps, and could even impair the learning of multi-agent cooperation. Predefined communication architectures, on the other hand, restrict communication among agents and thus restrain potential cooperation. To tackle these difficulties, in this paper, we propose an attentional communication model that learns when communication is needed and how to integrate shared information for cooperative decision making. Our model leads to efficient and effective communication for large-scale multi-agent cooperation. Empirically, we show the strength of our model in a variety of cooperative scenarios, where agents are able to develop more coordinated and sophisticated strategies than existing methods.

## 1 Introduction

Biologically, communication is closely related to and probably originated from cooperation. For example, vervet monkeys can make different vocalizations to warn other members of the group about different predators [3]. Similarly, communication can be crucially important in multi-agent reinforcement learning (MARL) for cooperation, especially for the scenarios where a large number of agents work in a collaborative way, such as autonomous vehicles planning [1], smart grid control [20], and multi-robot control [15].

Deep reinforcement learning (RL) has achieved remarkable success in a series of challenging problems, such as game playing [17, 22, 9] and robotics [13, 12, 6]. MARL can be simply seen as independent RL, where each learner treats the other agents as part of its environment. However, the strategies of other agents are uncertain and changing as training progresses, so the environment becomes unstable from the perspective of any individual agent and thus it is hard for agents to collaborate. Moreover, policies learned using independent RL can easily overfit to the other agents' policies [10].

We argue one of the keys to solve this problem is communication, which could enhance strategy coordination. There are several approaches for learning communication in MARL including DIAL [4], CommNet [23], BiCNet [19], and Master-Slave [8]. However, information sharing among all agents or in predefined communication architectures these methods adopt can be problematic. When there is a large number of agents, agents cannot differentiate valuable information that helps cooperative decision making from globally shared information, and hence communication barely helps and could even jeopardize the learning of cooperation. Moreover, in real-world applications, it is costly that all

---

agents communicate with each other, since receiving a large amount of information requires high bandwidth and incurs long delay and high computational complexity. Predefined communication architectures, *e.g.*, Master-Slave [8], might help, but they restrict communication among specific agents and thus restrain potential cooperation.

To tackle these difficulties, we propose an attentional communication model, called ATOC, to enable agents to learn effective and efficient communication under partially observable distributed environment for large-scale MARL. Inspired by recurrent models of visual attention, we design an attention unit that receives encoded local observation and action intention of an agent and determines whether the agent should communicate with other agents to cooperate in its observable field. If so, the agent, called *initiator*, selects collaborators to form a communication group for coordinated strategies. The communication group dynamically changes and retains only when necessary. We exploit a bidirectional LSTM unit as the communication channel to connect each agent within a communication group. The LSTM unit takes as input internal states (*i.e.*, encoding of local observation and action intention) and returns thoughts that guide agents for coordinated strategies. Unlike CommNet and BiCNet that perform arithmetic mean and weighted mean of internal states, respectively, our LSTM unit selectively outputs important information for cooperative decision making, which makes it possible for agents to learn coordinated strategies in dynamic communication environments.

We implement ATOC as an extension of actor-critic model, which is trained end-to-end by backpropagation. Since all agents share the parameters of the policy network, Q-network, attention unit, and communication channel, ATOC is suitable for large-scale multi-agent environments. We empirically show the success of ATOC in three scenarios, which correspond to the cooperation of agents for local reward, a shared global reward, and reward in competition, respectively. It is demonstrated ATOC agents are able to develop more coordinated and sophisticated strategies compared to existing methods. To the best of our knowledge, this is the first time that attentional communication is successfully applied to MARL.

## 2 Related Work

Recently, several models which are end-to-end trainable by backpropagation have been proven effective to learn communication in MARL. DIAL [4] is the first to propose learnable communication via backpropagation with deep Q-networks. At each timestep, an agent generates its message as the input of other agents for the next timestep. Gradients flow from one agent to another through the communication channel, bringing rich feedback to train an effective channel. However, the communication of DIAL is rather simple, just selecting predefined messages. Further, communication in terms of sequences of discrete symbols are investigated in [7] and [18].

CommNet [23] is a large feed-forward neural network that maps inputs of all agents to their actions, where each agent occupies a subset of units and additionally has access to a broadcasting communication channel to share information. At a single communication step, each agent sends its hidden state as the communication message to the channel. The averaged message from other agents is the input of next layer. However, it is only a large single network for all agents, so it cannot easily scale and would perform poorly in the environment with a large number of agents. It is worth mentioning that CommNet has been extended for abstractive summarization [2] in natural language processing.

BiCNet [19] is based on actor-critic model for continuous action, using recurrent networks to connect each individual agent's policy and value networks. BiCNet is able to handle real-time strategy games such as StarCraft micromanagement tasks. Master-Slave [8] is also a communication architecture for real-time strategy games, where the action of each slave agent is composed of contributions from both the slave agent and master agent. However, both works assume that agents know the global states of the environment, which is not realistic in practice. Moreover, predefined communication architectures restrict communication and hence restrain potential cooperation among agents. Therefore, they cannot adapt to the change of scenarios.

MADDPG [14] is an extension of actor-critic model for mixed cooperative-competitive environments. COMA [5] is proposed to solve multi-agent credit assignment in cooperative settings. MADDPG and COMA both use a centralized critic that takes as input the observations and actions of all agents. However, MADDPG and COMA have to train an independent policy network for each agent, where each agent would learn a policy specializing specific tasks [11], and the policy network easily overfits to the number of agents. Therefore, MADDPG and COMA are infeasible in large-scale MARL.

Mean Field [24] takes as input the observation and mean action of neighboring agents to make the decision. However, the mean action eliminates the difference among neighboring agents in terms of action and observation and thus incurs the loss of important information that could help cooperative decision making.

# 3 Background

**Deep Q-Networks (DQN).** Combining reinforcement learning with a class of deep neural networks, DQN [17] has performed at a level that is comparable to a professional game player. At each timestep $t$, the agent observes the state $s_t \in \mathcal{S}$, chooses an action $a_t \in \mathcal{A}$ according to the policy $\pi$, gets a reward $r_t$, and transitions to next state $s_{t+1}$. The objective is to maximize the total expected discounted reward $R_i = \sum_{t=0}^{T} \gamma^t r_t$, where $\gamma \in [0, 1]$ is a discount factor. DQN learns the action-value function $Q^\pi(s, a) = \mathbb{E}_s[R_t|s_t = s, a_t = a]$, which can be recursively rewritten as $Q^\pi(s, a) = \mathbb{E}_{s'}[r(s, a) + \gamma\mathbb{E}_{a'\sim\pi}[Q^\pi(s', a')]]$, by minimizing the loss: $\mathcal{L}(\theta) = \mathbb{E}_{s,a,r,s'}[y' - Q(s, a; \theta)]$, where $y' = r + \gamma\max_{a'} Q(s', a'; \theta)$. The agent selects the action that maximizes the Q value with a probability of $1 - \epsilon$ or acts randomly with a probability of $\epsilon$.

**Deterministic Policy Gradient (DPG).** Different from value-based algorithms like DQN, the main idea of policy gradient methods is to directly adjust the parameters $\theta$ of the policy to maximize the objective $J(\theta) = \mathbb{E}_{s\sim p^\pi, a\sim\pi_\theta}[R]$ along the direction of policy gradient $\nabla_\theta J(\theta)$, which can be written as $\nabla_\theta J(\theta) = \mathbb{E}_{s\sim p^\pi, a\sim\pi_\theta}[\nabla_\theta \log \pi_\theta(a|s) Q^\pi(s, a)]$. This can be further extended to deterministic policies [21] $\mu_\theta$: $\mathcal{S} \mapsto \mathcal{A}$, and $\nabla_\theta J(\theta) = \mathbb{E}_{s\sim\mathcal{D}}[\nabla_\theta\mu_\theta(a|s) \nabla_a Q^\mu(s, a)|_{a=\mu_\theta(s)}]$. To ensure $\nabla_a Q^\mu(s, a)$ exists, the action space must be continuous.

**Deep Deterministic Policy Gradient (DDPG).** DDPG [13] is an actor-critic algorithm based on DPG. It respectively uses deep neural networks parameterized by $\theta^\mu$ and $\theta^Q$ to approximate the deterministic policy $a = \mu(s|\theta^\mu)$ and action-value function $Q(s, a|\theta^Q)$. The policy network infers actions according to states, corresponding to the actor; the Q-network approximates the value function of state-action pair and provides the gradient, corresponding to the critic.

**Recurrent Attention Model (RAM).** In the process of perceiving the image, instead of processing the whole perception field, humans focus attention on some important parts to obtain information when and where it is needed and then move from one part to another. RAM [16] uses a RNN to model the attention mechanism. At each timestep, an agent obtains and processes a partial observation via a bandwidth-limited sensor. The glimpse feature extracted from the past observations is stored at an internal state which is encoded into the hidden layer of the RNN. By decoding the internal state, the agent decides the location of the sensor and the action interacting with the environment.

# 4 Methods

ATOC is instantiated as an extension of actor-critic model, but it can also be realized using value-based methods. ATOC consists of a policy network, a Q-network, an attention unit, and a communication channel, as illustrated in Figure 1.

We consider the partially observable distributed environment for MARL, where each agent $i$ receives a local observation $o_t^i$ correlated with the state $s_t$ at time $t$. The policy network takes the local observation as input and extracts a hidden layer as *thought*, which encodes both local observation and action intention, represented as $h_t^i = \mu_I(o_t^i; \theta^\mu)$. Every $T$ timesteps, the attention unit takes $h_t^i$ as input and determines whether communication is needed for cooperation in its observable field. If needed, the agent, called *initiator*, selects other agents, called *collaborators*, in its field to form a communication group and the group stays the same in $T$ timesteps. Communication is fully determined (when and how long to communicate) by the attention unit when $T$ is equal to 1. $T$ can also be tuned for the consistency of cooperation. The communication channel connects each agent of the communication group, takes as input the thought of each agent and outputs the integrated thought that guides agents to generate coordinated actions. The integrated thought $\tilde{h}_t^i$ is merged with $h_t^i$ and fed into the rest of the policy network. Then, the policy network outputs the action $a_t^i = \mu_{II}(h_t^i, \tilde{h}_t^i; \theta^\mu)$. By sharing encoding of local observation and action intention within a dynamically formed group, individual agents could build up relatively more global perception of the environment, infer the intent of other agents, and cooperate on decision making.

### 4.1 Attention model

When the coach directs a team, instead of managing a whole scene, she focuses attention selectively on the key position and gives some directional but not specific instructions to the players near the location. Inspired from this, we introduce the attention mechanism to learning multi-agent communication. Different from the coach, our attention unit never senses the environment in full, but only uses encoding of observable field and action intention of an agent and decides whether communication is helpful in terms of cooperation. The attention unit can be instantiated by RNN or MLP. The first part of the actor network that produces the thought corresponds to the glimpse network and the thought $h_t^i$ can be considered as the glimpse feature vector. The attention unit takes the thought representation as input and produces the probability of the observable field of the agent becomes an attention focus (*i.e.*, the probability of communication).

Unlike existing work on learning communication in MARL, *e.g.*, CommNet and BiCNet, where all agents communicate with each other all the time, our attention unit enables dynamic communication among agents only when necessary. This is much more practical, because in real-world applications communication is restricted by bandwidth and/or range and incurs additional cost, and thus it may not be possible or cost too much to maintain full connectivity among all the agents. On the other hand, dynamic communication can keep the agent from receiving useless information compared to full connectivity. As will be discussed in next section, useless information may negatively impact cooperative decision making among agents. Overall, the attention unit leads to more effective and efficient communication.

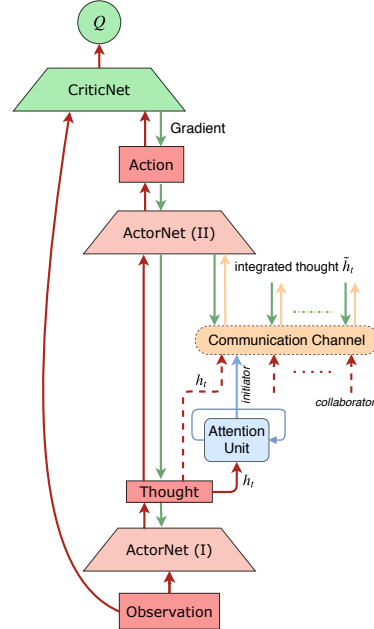

Figure 1: ATOC architecture.

### 4.2 Communication

When an initiator selects its collaborators, it only considers the agents in its observable field and ignores those who cannot be perceived. This setting complies with the facts: (*i*) one of the purposes of communication is to share the partial observation, and adjacent agents could understand each other easily; (*ii*) cooperative decision making can be more easily accomplished among adjacent agents; (*iii*) all agents share one policy network, which means adjacent agents may have similar behaviors, however communication can increase the diversity of their strategies. There are three types of agents in the observable field of the initiator: other initiators; agents who have been selected by other initiators; agents who have not been selected. We assume a fixed communication bandwidth, which means each initiator can select at most $m$ collaborators. The initiator first chooses collaborators from agents who have not been selected, then from agents selected by other initiators, finally from other initiators, all based on proximity.

When an agent is selected by multiple initiators, it will participate the communication of each group. Assuming agent $k$ is selected by two initiators $p$ and $q$ sequentially. Agent $k$ first participates in the communication of $p$'s group. The communication channel integrates their thoughts: $\{\tilde{h}_t^p, \cdots, \tilde{h}_t^{k'}\} = g(h_t^p, \cdots, h_t^k)$. Then agent $k$ communicates with $q$'s group: $\{\tilde{h}_t^q, \cdots, \tilde{h}_t^{k''}\} = g(h_t^q, \cdots, h_t^{k'})$. The agent shared by multiple groups bridges the information gap and strategy division among individual groups. It can disseminate the thought within a group to other groups, which can eventually lead to coordinated strategies among the groups. This is especially critical for the case where all agents collaborate on a single task. In addition, to deal with the issue of role assignment and heterogeneous agent types, we can fix the position of agents who participate in communication.

The bi-directional LSTM unit acts as the communication channel. It plays the role of integrating internal states of agents within a group and guiding the agents towards coordinated decision making. Unlike CommNet and BiCNet that integrate shared information of agents by arithmetic mean and weighted mean, respectively, our LSTM unit can selectively output information that promotes cooperation and forget information that impedes cooperation through gates.

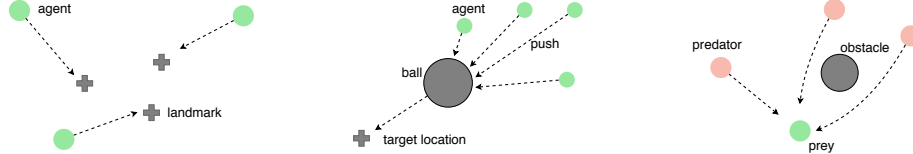

Figure 2: Illustration of experimental scenarios: *cooperative navigation* (left), *cooperative pushball* (mid), *predator-prey* (right).

## 4.3 Training

The training of ATOC is an extension of DDPG. More concretely, consider a game with $N$ agents, and the critic, actor, communication channel, and attention unit of ATOC is parameterized by $\theta^Q$, $\theta^\mu$, $\theta^g$, and $\theta^p$, respectively. Note that we drop time $t$ in the following notations for simplicity. The experience replay buffer $\mathcal{R}$ contains the tuples $(O, A, R, O', C)$ recording the experiences of all agents, where $O = (o_1, \ldots, o_N)$, $A = (a_1, \ldots, a_N)$, $R = (r_1, \ldots, r_N)$, $O' = (o'_1, \ldots, o'_N)$, and $C$ is a $N \times N$ matrix that records the communication groups. We select experiences where the action is determined by an agent independently (*i.e.*, without communication) and experiences with communication, respectively, to update the action-value function $Q^\mu$ as:

$$\mathcal{L}(\theta^Q) = \mathbb{E}_{o,a,r,o'}\left[(Q^\mu(o,a) - y)^2\right], \quad y = r + \gamma Q^{\mu'}(o',a')\,|_{a'=\mu'(o')}.$$

The policy gradient can be written as:

$$\nabla_{\theta^\mu} J(\theta^\mu) = \mathbb{E}_{o,a\sim\mathcal{R}}\left[\nabla_{\theta^\mu}\mu(a|o)\,\nabla_a Q^\mu(o,a)\,|_{a=\mu(o)}\right].$$

By the chain rule, the gradient of integrated thought can be further derived as:

$$\nabla_{\theta^g} J(\theta^g) = \mathbb{E}_{o,a\sim\mathcal{R}}\left[\nabla_{\theta^g} g(\tilde{h}|H)\nabla_{\tilde{h}}\mu(a|\tilde{h})\nabla_a Q^\mu(o,a)\,|_{a=\mu(o)}\right].$$

The gradients are backpropagated to the policy network and communication channel to update the parameters. Then, we softly update target networks as $\theta' = \tau\theta + (1-\tau)\theta'$.

The attention unit is trained as a binary classifier for communication. For each initiator $i$ and its group $G_i$, we calculate the difference of mean Q values between coordinated actions and independent actions (denoted as $\bar{a}$)

$$\Delta Q_i = \frac{1}{|G_i|}\left(\sum_{j\in G_i} Q\left(o_j, a_j | \theta^Q\right) - \sum_{j\in G_i} Q\left(o_j, \bar{a}_j | \theta^Q\right)\right)$$

and store $(\Delta Q_i, h^i)$ into a queue $\mathcal{D}$, where $\Delta Q$ weights the performance enhancement produced by communication. When an episode ends, we perform min-max normalization on $\Delta Q$ in $\mathcal{D}$ and get $\Delta \hat{Q} \in [0,1]$. $\Delta \hat{Q}$ can be used as the tag of the binary classifier and we use log loss to update $\theta^p$ as:

$$\mathcal{L}(\theta^p) = -\Delta \hat{Q}_i \log(p\left(h^i|\theta^p\right)) - (1-\Delta \hat{Q}_i)\log\left(1 - p\left(h^i|\theta^p\right)\right).$$

## 5 Experiments

Experiments are performed based the multi-agent particle environment [14, 18], which is a two-dimensional world with continuous space and discrete time, consisting agents and landmarks. We made a few modifications to the environment so as to adopt a large number of agents. Each agent has only local observation, acts independently and cooperatively, and collects its own reward or a shared global reward. We perform experiments in three scenarios, as illustrated in Figure 2, to investigate the cooperation of agents for local reward, shared global reward and reward in competition, respectively. We compare ATOC with CommNet, BiCNet and DDPG. CommNet and BiCNet are the full communication model, and DDPG is exactly ATOC without communication. MADDPG has to train an independent policy network for each agent, which makes it infeasible in large-scale MARL.

### 5.1 Hyperparameters

In all the experiments, we use Adam optimizer with a learning rate of 0.001. The discount factor of reward $\gamma$ is 0.96. For the soft update of target networks, we use $\tau = 0.001$. The neural networks use

ReLU and batch normalization for some hidden layers. The actor network has four hidden layers, the second layer is the thought (128 units), and the output layer is the $\tanh$ activation function. The critic network has two hidden layers with 512 and 256 units respectively. We use two-layer MLP to implement the attention unit but it is also can be realized by RNN. For communication, $T$ is 15. We initialize all of the parameters by the method of *random normal*. The capacity of the replay buffer is $10^5$ and every time we take a minibatch of 2560. We noted that the large minibatch can accelerate the convergence process, especially for the case of sparse reward. We accumulate experiences in the first thirty episodes before training. As DDPG, we use an Ornstein-Uhlenbeck process with $\theta = 0.15$ and $\sigma = 0.2$ for the exploration noise process.

## 5.2 Cooperative Navigation

In this scenario, $N$ agents cooperatively reach $L$ landmarks, while avoiding collisions. Each agent is rewarded based on the proximity to the nearest landmark, while it is penalized when colliding with other agents. Ideally, each agent predicts actions of nearby agents based on its own observation and received information from other agents, and determines its own action towards occupying a landmark without colliding with other agents.

We trained ATOC and the baselines with the settings of $N = 50$ and $L = 50$, where each agent can observe three nearest agents and four landmarks with relative positions and velocities. At each timestep, the reward of an agent is $-d$, where $d$ denotes the distance between the agent and its nearest landmark, or $-d - 1$ if a collision occurs. Figure 3 shows the learning curves of 3000 episodes in terms of mean reward, averaged over all agents and timesteps. We can see that ATOC converges to higher mean reward than the baselines. We evaluate ATOC and the baselines by running 30 test games and measure average mean reward, number of collisions, and percentage of occupied landmarks.

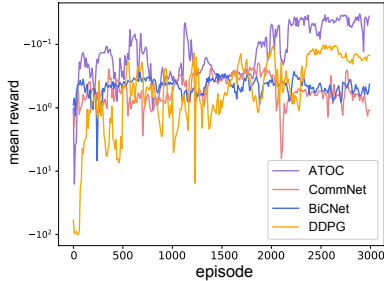

Figure 3: Reward of ATOC against baselines during training on cooperative navigation.

As shown in Table 1, ATOC largely outperforms all the baselines. In the experiment, CommNet, BiCNet and DDPG all fail to learn the strategy that ATOC obtains. That is an agent is first trying to occupy the nearest landmark. If the landmark is more likely to be occupied by other agent, the agent will turn to another vacant landmark rather than keeping probing and approaching the nearest landmark. The strategy of DDPG is more aggressive, *i.e.*, multiple agents usually approach a landmark simultaneously, which could lead to collisions. Both CommNet and BiCNet agents are more conservative, *i.e.*, they are more willing to avoid collisions rather than seizing a landmark, which eventually leads to a small number of occupied landmarks. Moreover, both CommNet and BiCNet agents are more likely to surround a landmark and observe the actions of other agents. Nevertheless, gathered agents are prone to collisions.

As ATOC without communication is exactly DDPG and ATOC outperforms DDPG, we can see communication indeed helps. However, CommNet and BiCNet also have communication, *why is their performance much worse?* CommNet performs arithmetic mean on the information of the hidden layers. This operation implicitly treats information from different agents equally. However, information from various agents has different value for an agent to make decisions. For example, the information from a nearby agent who intends to seize the same landmark is much more useful than the information from an agent far away. In the scenario with a large number of agents, there is a lot of useless information, which can be seen as noise that interferes the decision of agents. BiCNet uses

Table 1: Cooperative Navigation

|  |  | ATOC | ATOC w/o Comm. | DDPG | CommNet | BiCNet |
|---|---|---|---|---|---|---|
| $N = 50, L = 50$ | mean reward | $-0.04$ | $-0.22$ | $-0.14$ | $-0.60$ | $-0.52$ |
|  | # collisions | 13 | 47 | 32 | 59 | 51 |
|  | % occupied landmarks | 92% | 40% | 22% | 12% | 16% |
| $N = 100, L = 100$ | mean reward | $-0.05$ |  | $-0.23$ | $-0.65$ | $-0.73$ |
|  | # collisions | 28 |  | 53 | 68 | 91 |
|  | % occupied landmarks | 89% |  | 25% | 17% | 9% |

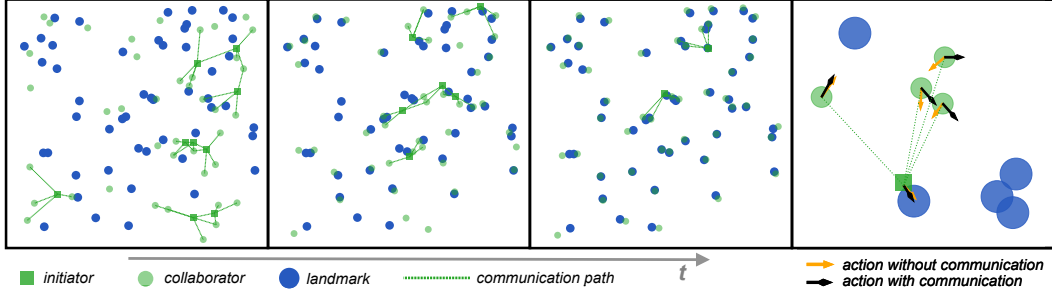

*initiator* ■   *collaborator* ●   *landmark* ●   ·········· *communication path*   $t$ →   → *action without communication*   → *action with communication*

Figure 4: Visualizations of communications among ATOC agents on cooperative navigation. The rightmost figure illustrates actions taken by a group of agents with and without communication.

a RNN as the communication channel, which can be seen as the weighted mean. However, as the number of agents increases, RNN also fails to capture the importance of information from different agents. Unlike CommNet and BiCNet, ATOC exploits the attention unit to dynamically perform communication, and most information is from nearby agents and thus helpful for decision making.

It is essential for agents to share a policy network as they do in the experiments. The primary reason is that most real-world applications are open systems, *i.e.*, agents come and go. If each agent is trained with an independent policy network, the network is apt to overfit to the number of agents in the environment and thus hard to generalize, not to mention the efforts needed to train numerous independent policy networks, like MADDPG, in large-scale multi-agent environments. However, agents that share a policy network may be homogeneous in terms of strategy, *e.g.*, DDPG agents are all aggressive to seize the landmarks while CommNet and BiCNet agents are all conservative. Nevertheless, unlike these baselines, ATOC agents behave differently: when a landmark is more likely be occupied by an agent, nearby agents will turn to other landmarks. The primary reason behind this is the communication scheme of ATOC. An agent can share its local observation and intent to nearby agents, *i.e.*, the dynamically formed communication group. Although the size of communication group is small, the shared information may be further encoded and forwarded among groups by the agent who belongs to multiple groups. Thus, each agent can obtain more and diverse information. Based on the received information, agents may infer the actions of other agents and behave accordingly. Overall, ATOC agents show cooperative strategies to occupy the landmarks.

To investigate the scalability of ATOC and the baselines, we directly use the trained models under the setting of $N = 50$ and $L = 50$ to the scenario of $N = 100$ and $L = 100$. With the increase of agent density, the number of collisions of all the methods increases. However, as shown in Table 1, ATOC is still much better than the baselines in terms of all the metrics, which proves the scalability of ATOC. Interestingly, the percentage of occupied landmarks increases for DDPG and CommNet. As discussed before, the learned strategy of CommNet is conservative in the original setting, and thus it might lead to more occupied landmarks when agents are dense and decisions are more conflicting. The percentage of occupied landmarks of DDPG increases slightly, the number of collisions increase though. The largely degraded performance of BiCNet in terms of all the metrics shows its bad scalability.

We visualize the communications among ATOC agents to trace the effect of the attention unit. As illustrated in Figure 4 (the left three figures), attentional communications occur at the regions where agents are dense and situations are complex. As the game progresses, the agents occupy more landmarks and communication is less needed. We select a communication group and observe their behaviors with/without communications. We find that agents without communications are more likely to target the same landmarks, which may lead to collisions, while agents with communications can spread to different landmarks, as depicted in Figure 4 (the rightmost figure).

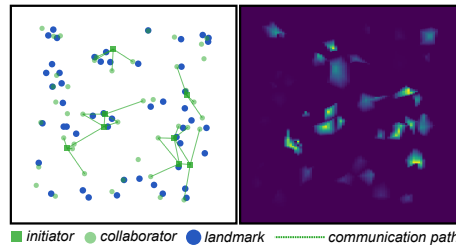

*initiator* ■   *collaborator* ●   *landmark* ●   ·········· *communication path*

Figure 5: Heatmap of attention corresponding to communication among ATOC agents in cooperative navigation.

To investigate the correlation between communication and attention, we further visualize the communication among ATOC agent at certain timestep and its corresponding heatmap of attention in

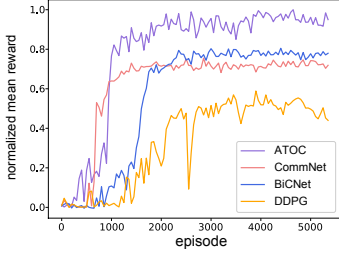

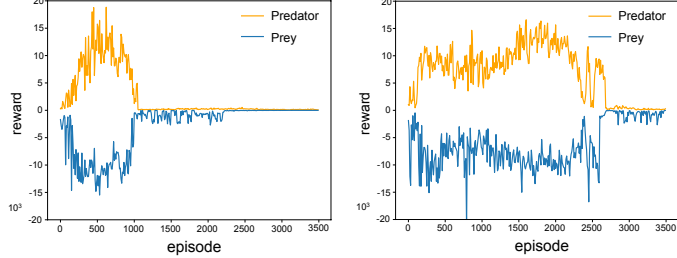

Figure 6: Reward of ATOC against baselines during training on cooperative pushball.

Figure 7: Learning curves of ATOC (left) and CommNet (right) during learning on predator-prey.

Figure 5. The regions where communications occur are the attention focuses as illustrated in Figure 5. Only at the regions where agents are dense and landmarks are not occupied, communication is needed for cooperative decision making. Our attention unit learns exactly what we expect, *i.e.*, carrying out communication only when needed. Further, we turn off the communications of ATOC agents (without retraining) and the performance drops as shown in Table 1. Therefore, we argue that communication during execution is also essential for better cooperation.

## 5.3 Cooperative Pushball

In this scenario, $N$ agents who share a global reward cooperatively push a heavy ball to a designated location. The ball is 200 times heavier and 144 times bigger than an agent. Agents push the ball by collisions, not by forces, and control the moving direction by hitting the ball at different angles. However, agents are not given the prior knowledge of how to control the direction, which is learned during training. The inertial mass of the ball makes it difficult for agents to change its state of motion, and round surfaces of the ball and agents make the task more complicated. Therefore, the task is very challenging. In the experiment, there are 50 agents, each agent can observe the relative locations of the ball and at most 10 agents within a predefined distance, and the designated location is the center of the playground. The reward of agents at each timestep is $-d$, where $d$ denotes the distance from the ball to the center of the playground.

Figure 6 shows the learning curve in terms of normalized mean reward for ATOC and the baselines. ATOC converges to a much higher reward than all the baselines. CommNet and BiCNet have comparable reward, which is higher than DDPG. We evaluate ATOC and the baselines by running 30 test games. The normalized mean reward is illustrated in Table 2.

ATOC agents learn sophisticated strategies: agents push the ball by hitting the center of the ball; they change the moving direction by striking the side of the ball; when the ball is approaching the target location, some agents will turn to the opposite of moving direction of the ball and collide with the ball to reduce its speed so as to keep the ball from passing the target location; at the end agents split into two parts with equal size and strike the ball from two opposite directions, and eventually the ball will be stabilized at the target location. The control of moving direction and reducing speed embodies the division of work and cooperation among agents, which is accomplished by communication. By visualizing the communication structures and behaviors of agents, we find that agents in the same communication group behave homogeneously, *e.g.*, a group of agents push the ball, a group of agents control the direction, and a group of agents reduce the speed when the ball approaches the target location.

DDPG agents all behave similarly and show no division of work. That is almost all agents push the ball from the same direction, which can lead to the deviated direction or quickly passing the target location. Until the ball is pushed far from the target location, DDPG agents realize they are pushing at the wrong direction and switch to the opposite together. Therefore, the ball is pushed back and force and hardly stabilized at the target location. Communication indeed helps, which explains why CommNet and BiCNet are better than DDPG. ATOC is better than CommNet and BiCNet, which is

Table 2: Cooperative Pushball

|  | ATOC | ATOC w/o Comm. | DDPG | CommNet | BiCNet |
|---|---|---|---|---|---|
| normalized mean reward | 0.95 | 0.86 | 0.50 | 0.71 | 0.77 |

reflected in the experiments by ATOC's much smaller amplitude of oscillation. The primary reason has been explained in previous section.

To investigate the effect of communication in ATOC, we turn off the communication of ATOC agents (without retraining), and the result is shown in Table 2. The performance of ATOC drops, but it is still better than all the baselines. The reason behind this is that communication stabilizes the environment during training. Moreover, in ATOC, *cooperative policy gradients* can backpropagate to update individual policy networks, which enables agents to infer the actions of other agents without communication and thus behaves cooperatively.

## 5.4 Predator-Prey

In this variant of the classic predator-prey game, 60 slower predators chase 20 faster preys around an environment with 5 landmarks impeding the way. Each time a predator collides with an prey, the predator is rewarded with $+10$ while the prey is penalized with $-10$. Each agent observes the relative positions and velocities of five nearest predators and three nearest preys, and the positions of two nearest landmarks. To restrain preys in the playground instead of runaway, a prey is also given a reward based on its coordinates $(x, y)$ at each timestep. The reward is $-f(x) - f(y)$, where $f(a) = 0$ if $a \leq 0.9$, $f(a) = 10 \times (a - 0.9)$ if $0.9 < a < 1$, otherwise $f(a) = e^{2a-2}$.

In this scenario, predators collaborate to surround and seize preys, while preys cooperate to perform temptation and evasion. In the experiment, we focus on the cooperation of predators/preys rather than the competition between them. For each method, predator and prey agents are trained together. Figure 7 shows the learning curves of predators and preys for ATOC and CommNet. As the learning curves of DDPG and BiCNet are not stable in this scenario, we only show the

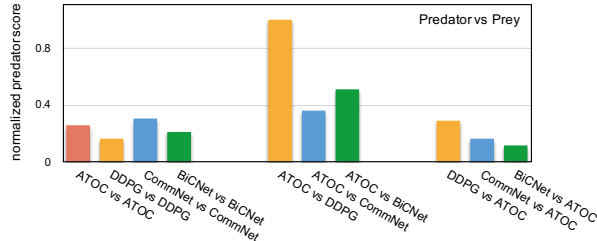

Figure 8: Cross-comparison between ATOC and baselines in terms of predator score on predator-prey.

learning curves of ATOC and CommNet. From Figure 7, we can see that ATOC converges much faster than CommNet, where ATOC is stabilized after 1000 episodes, but CommNet is stabilized after 2500 episodes. As the setting of the scenario appears to be more favorable for predators than preys, which is also indicated in Figure 7, both ATOC and CommNet predators are converged more quickly than preys.

To evaluate the performance, we perform cross-comparison between ATOC and the baselines. That is we play the game using ATOC predators against preys of the baselines and vice versa. The results are shown in terms of the 0-1 normalized mean predator score of 30 test runs for each game, as illustrated in Figure 8. The first bar cluster shows the games between predators and preys of the same method, from which we can see that the game setting is indeed more favorable for predators than preys since predators have positive scores for all the methods. The second bar cluster shows the scores of the games where ATOC predators are against DDPG, CommNet, and BiCNet preys. We can see that ATOC predators have higher scores than the predators of all the baselines and hence are stronger than other predators. The third bar cluster shows the games where DDPG, CommNet, and BiCNet predators are against ATOC preys. The predator scores are all low, comparable to the scores in the first cluster. Therefore, we argue that ATOC leads to better cooperation than the baselines even in competitive environments and the learned policy of ATOC predators and preys can generalize to the opponents with different policies.

## 6 Conclusions

We have proposed an attentional communication model in large-scale multi-agent environments, where agents learn an attention unit that dynamically determines whether communication is needed for cooperation and also a bi-directional LSTM unit as a communication channel to interpret encoded information from other agents. Unlike existing methods for communication, ATOC can effectively and efficiently exploits communication to make cooperative decisions. Empirically, ATOC outperforms existing methods in a variety of cooperative multi-agent environments.

**Acknowledgments**

This work was supported in part by Peng Cheng Laboratory and NSFC under grant 61872009.

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
