[Supplementary Material · ATOC_appendix.pdf]

# Appendix

## Attentional Communication Algorithm

For completeness, we provide the ATOC algorithm as below.

---

**Algorithm 1** Attentional communication

---

1: Initialize critic network, actor network, communication channel and attention classifier
   Initialize target networks
   Initialize replay buffer $\mathcal{R}$
   Initialize queue $\mathcal{D}$
2: **for** episode = $1, \ldots, \mathcal{M}$ **do**
3:     **for** $t = 1, \ldots, \mathcal{T}$ **do**
4:         Get thought $h_t^i = \pi_\text{I}\left(o_t^i; \theta^\mu\right)$ for each agent $i$
5:         Each agent $i$ decides whether to initiate communication based on $h_i$ every $T$ timesteps
6:         **for** $i$ in initiators **do**
7:           $(\tilde{h}_t^i, \tilde{h}_t^1, \ldots, \tilde{h}_t^j) = g(h_t^i, h_t^1, \ldots, h_t^j; \theta^g)$, where agent 1 to $j$ in $i$'s group
8:         **end for**
9:         Select action $a_t^i = \pi_\text{II}(h_t^i, \tilde{h}_t^i; \theta^\mu) + \mathcal{N}_t^i$ for each agent $i$ with communication
10:       Select action $a_t^i = \pi_\text{II}(h_t^i; \theta^\mu) + \mathcal{N}_t^i$ for each agent $i$ without communication
11:       Execute action $a_t^i$, obtain reward $r_t^i$, and get new observation $o_{t+1}^i$ for each agent $i$
12:       **for** $i$ in initiators **do**
13:         Get action $\bar{a}_t^j = \pi_\text{II}(h_t^j; \theta^\mu)$ for each agent $j$ in $i$'s group $G_i$
14:         Compute difference of mean Q values with and without communication:
$$\Delta Q_t^i = \frac{1}{|G_i|} \left( \sum_{j \in G_i} Q\left(o_t^j, a_t^j | \theta^Q\right) - \sum_{j \in G_i} Q\left(o_t^j, \bar{a}_t^j | \theta^Q\right) \right)$$
15:         Store $\left(h_t^i, \Delta Q_t^i\right)$ in $\mathcal{D}$
16:       **end for**
17:       Store transition $(O_t, A_t, R_t, O_{t+1}, C_t)$ in $\mathcal{R}$
18:       Sample a random minibatch of $N$ transitions from $\mathcal{R}$
19:       Sample agents without communication
           Update the critic $\theta^Q$ by minimizing $\mathcal{L}(\theta^Q)$
           Update the actor $\theta^\mu$ using sampled policy gradients $\nabla_{\theta^\mu} J\left(\theta^\mu\right)$
20:       Sample agents with communication
           Update the critic $\theta^Q$ by minimizing $\mathcal{L}(\theta^Q)$
           Update the actor $\theta^\mu$ using sampled policy gradients $\nabla_{\theta^\mu} J\left(\theta^\mu\right)$
           Update the communication channel $\theta^g$ using sampled thoughts gradients $\nabla_{\theta^g} J\left(\theta^g\right)$
21:       Update the target networks: $\theta' = \tau\theta + (1 - \tau)\theta'$
22:     **end for**
23:     Get $\Delta \hat{Q}_i \in [0, 1]$ by min-max normalization for each $\Delta Q_i$ in $\mathcal{D}$
24:     Update the attention classifier $\theta^p$ by minimizing log loss:
$$\mathcal{L}(\theta^p) = -\Delta \hat{Q}_i \log(p\left(h^i | \theta^p\right)) - (1 - \Delta \hat{Q}_i) \log\left(1 - p\left(h^i | \theta^p\right)\right)$$
25:     Empty the queue $\mathcal{D}$
26: **end for**

---

## Video

The video of the experiments is given by this link <https://goo.gl/X8wBqw>. The video shows the games of cooperative navigation and cooperative pushball. As the performance of different methods cannot be visually differentiated in predator-prey, predator-prey is not included. The video aims to highlight different behaviors of agents trained using different methods. So, we show the games of ATOC, DDPG, and ComNet agents in cooperative navigation and ATOC and DDPG agents in cooperative pushball.