[Reviews · NeurIPS 2018]

Reviewer 1



Main Ideas: -- They present a multi-agent RL framerork in which the agents choose when and who to colloborate based on a new attention network structure. -- This eliminates excessive communication among agents, as well as provides dynamic communication schema, which is less restrictive. Strengths: -- The communication is selective based on the attention and only considers the nearby agents, without considering unnessary communication with distant agents. This type of learning fits to games scnearoies like finding lanmarks in a maze where muliple communative agents will be helpful. -- The communication model tested on three different tasks, to show the strenght of the new approach. -- The paper is well written, organized and motivation is clear. Weaknesses: -- Only toy tasks are used. -- The same network with no communication or communication with all the rest of the agents should be two of the baselines of this model. -- The experiments section investigates different episodes of the model yielding insteresting conclusions about the fact that when communication is necessary at first but later less communication is enough at later stages when the agents do reach the goal states. This is expected results but i would like to know if the improvement in the results can be attributed to the attention that enables dynamic communication among agents, or the DDPG or any other component they used to improve the performance. These ablations should be added to the paper. Questions: -- The folloowing paper also uses hiearchial attention over multiple agents missing from the reference list: 'Deep Communicating Agents for Abstractive Summarization' Celikyilmaz et.al., NAACL 2018 Fix: -- Line 10 : 'how to integrates' -- Line 30-31 : Please indicate shortly what are these other communication models such as DIAL, etc. rather than listing their names, which does not make much sense.

Reviewer 2



Summary: This paper describes and evaluates a complex reinforcement learning algorithm for achieving cooperation within a team of agents. As I understand it, in this design, the agents all share the same policy network, but they sense different things. To help the agents coordinate their behavior, the system learns when to communicate and combine “thoughts” (outputs from a layer of a neural network) with each other. The authors empirically analyze the system in three different domains (cooperative navigation, cooperative push ball, and Predator-Prey). The results achieved appear to outperform existing algorithms designed for similar situations. Quality: The authors appear to combine together state-of-the-art algorithmic mechanisms in a very clever way. The results suggest that the algorithm is effective. I very much like that the authors evaluated the system in three different domains. These attributes make the paper quite appealing. But no paper is perfect (at least from the eye of reviewers). In my opinion, the quality of the paper could be improved in three ways: 1. Provide more details in order to make the paper reproducible: To me, the algorithm and results do not appear to be reproducible given the descriptions provided in the paper. While the paper does a good job of overviewing the system, important details (parameter values, network designs, etc.) that are needed to reproduce the results are not given (at least that I saw). I believe that it is likely that someone else trying to implement similar mechanisms but without the same details would get very different results (particularly if there were not experts in the field), as I believe these details are likely critical to the success of the system. That said, I realize that providing sufficient details to reproduce complex algorithms is very difficult in a short paper format. 2. More thorough algorithmic analysis: From the analysis of the results, it is a bit difficult to determine what makes the system tick. Comparisons to previously developed algorithms are nice to establish baselines, but do not thoroughly isolate in all instances what makes the algorithms tick. The authors do add nice analysis (e.g., testing ATOC with and without communication). But even without communication, ATOC outperforms the other algorithms substantially in cooperative pushball (Table 2). Is this because the learning mechanisms used in ATOC are just better, or did the authors just take more time to refine and optimize ATOC for the given domains more than the other algorithms? 3. Statistical comparisons: It is also somewhat difficult to interpret the differences between the various algorithms, as no statistical tests or error bars are provided to communicate to the reader significance. Clarity: The organization of the paper is quite good, and it is definitely possible for a reader to get a good high-level understanding of what is being done. I do have a few suggestions (nitpicks in some sense) that may be helpful in providing additional clarity. 1. At least from my perspective, the abstract and introduction didn’t spell out the domain very well up front. Agents that “cooperate” with each other can take on many forms. The algorithmic architecture described herein addresses one of these forms, that of a team of computer agents that share many aspects of the same “brain.”. As recommended by Shoham et. al, 2007 (If multi-learning is the answer, what is the question), I recommend better spelling this out from the onset. The domain addressed in this work (and hence potentially the kinds of algorithms that will work) is very different from other recent and older work in which communication of AI systems is designed and evaluated, including systems that interact with people and systems in which cooperation can be beneficial but in which the agents do not share the same interests. 2. I may have missed it, but a good description of how the rewards are computed for each domain appears to be missing. For example, I’m not sure how good a reward of -0.04 is (or what it represents) in the “cooperative navigation” task (Table 1). 3. Several times throughout the paper, the authors use the terms “barely help,” “barely adapt,” and “hardly adapts” to describe other algorithms. Each time these terms are used, I got confused. The language seemed to be imprecise and could be interpreted as just attempts to “beat up” previous work. I believe the authors are trying to communicate that the previous algorithms can sometimes have issues “scaling up” because of various design decisions — I just recommend using different language to describe that. Originality: The work builds on and appears to improve upon a line of work that is emerging in the recent literature: communication in team settings in which a single designer controls all robots. I believe that the algorithmic architecture is novel. Significance: On one hand, the authors appear to be combining together advanced methods in a very clever way. The results achieved are relatively good, and it does appear that the work advances the state-of-the-art. On the other hand, the perspective is a little bit strange, neither using a true multi-agent system (wherein all agents are separate) nor a fully centralized system. Since the designer appears to have full control of the algorithms used by all robots, is there a reason that a computationally adept, but fully centralized system could not be used for computation rather than a “multi-agent system.” Overall, the assumptions of the paper are not fully spelled out, nor are the algorithmic mechanisms thoroughly evaluated in isolation. Too me, this dampens the significance of the contributions, but certainly does not eliminate them. I find this paper to be a nice work.

Reviewer 3



Answer to rebuttal Given your response (and most importantly "DDPG is exactly ATOC without communication") I have increased the score one point. Please make sure to make this more clear in the final version. Also, make sure to include the reward function. Learning Attentional Communication for Multi-Agent Cooperation The paper address the problem of multi-agent cooperation with focus on local inter agent communication. Hence, apart from taking actions affecting the environment the agents are able to share their state with a group of m other agent. However, as I understand the paper, the communication is not as one might expect a one to many broadcasting of information but rather a mind meld where all participating agents states are integrated using an LSTM and fed back to all agents in the group. These communication groups are formed using input from the individual agents “Attention Unit” indicating if each respective agent is interested in starting a state sharing group. If at least one agent is interested a group will be created and populated based on proximity (and if the agents are already part of another group). However, there is no way for an agent to opt out. Further, to stabilize training all agents share weights which bypasses the problem of non stationarity usually associated with MARL and also makes it computationally easy to scale up with more agents. To validate the performance of their proposed method three experiments of well known multi agent problems have been run and compared to baselines and two SOTA systems. Comments: § It is interesting that you get good results even when communication is turned off during testing (but on during training). You attribute this to the communication helping with training but provide no evidence for this. It would be nice to see results where communication was off both during training and testing for comparison. § I don’t see why you are referring to the RNN that decides whether to initiate state sharing as an “Attention Unit” since, as far as I can see, it has no impact on the agents own input. § You employ a lot of analogies and examples in the text which is great to help understanding, however, I sometimes found the text lacking in details regarding the algorithm, e.g. I was not able to find the actual reward functions used in the experiments. § Are figures 3 and 5 showing data from one training instance or the mean over many runs? If it is the former I would be hesitant to draw any conclusions from it and if it is the latter you should make sure to indicate that in the paper along with the number of runs you have computed your average over. Minor comments: § Though the text is reasonably well structured grammatical problems throughout the text makes it hard to read something that would need to be addressed if the paper were to be accepted. § in section 5.2 you refer to figure 3 though I think you you actually mean to refer to fig 5.